# Renal Tissue miRNA Expression Profiles in ANCA-Associated Vasculitis—A Comparative Analysis

**DOI:** 10.3390/ijms23010105

**Published:** 2021-12-22

**Authors:** Matic Bošnjak, Željka Večerić-Haler, Emanuela Boštjančič, Nika Kojc

**Affiliations:** 1Faculty of Medicine, Institute of Pathology, University of Ljubljana, Korytkova Ulica 2, 1000 Ljubljana, Slovenia; matic.bosnjak@mf.uni-lj.si; 2Faculty of Medicine, University of Ljubljana, Vrazov Trg 2, 1000 Ljubljana, Slovenia; zeljka.vecerichaler@kclj.si; 3Department of Nephrology, University Clinical Centre Ljubljana, Zaloška Cesta 7, 1000 Ljubljana, Slovenia

**Keywords:** microRNA, screening, ANCA, vasculitis, glomerulonephritis, epigenetics, pathogenesis, biomarker, renal disease

## Abstract

Anti-neutrophil cytoplasm antibody (ANCA)-associated vasculitis (AAV) comprises autoimmune disease entities that cause target organ damage due to relapsing-remitting small vessel necrotizing vasculitis, and which affects various vascular beds. The pathogenesis of AAV is incompletely understood, which translates to considerable disease- and treatment-related morbidity and mortality. Recent advances have implicated microRNAs (miRNAs) in AAV; however, their accurate characterization in renal tissue is lacking. The goal of this study was to identify the intrarenal miRNA expression profile in AAV relative to healthy, non-inflammatory and inflammatory controls to identify candidate-specific miRNAs. Formalin-fixed, paraffin-embedded renal biopsy tissue samples from 85 patients were obtained. Comprehensive miRNA expression profiles were performed using panels with 752 miRNAs and revealed 17 miRNA that differentiated AAV from both controls. Identified miRNAs were annotated to characterize their involvement in pathways and to define their targets. A considerable subset of differentially expressed miRNAs was related to macrophage and lymphocyte polarization and cytokines previously deemed important in AAV pathogenesis, lending credence to the obtained results. Interestingly, several members of the miR-30 family were detected. However, a validation study of these differentially expressed miRNAs in an independent, larger sample cohort is needed to establish their potential diagnostic utility.

## 1. Introduction

The term anti-neutrophil cytoplasm antibody (ANCA)-associated vasculitis (AAV) encompasses a group of rare, pathogenically distinct entities that share autoimmune small vessel necrotizing vasculitis as a prominent feature, a condition which results in severe target organ damage. As a heterogeneous disease group, AAV is currently best classified by the presence (or absence) of specific serum ANCAs that serve as standard diagnostic and disease activity markers, with myeloperoxidase (MPO) and proteinase-3 (PR3) as the two principal antigens that pathogenic ANCAs target. Consequently, AAV is termed either PR3-ANCA-positive (PR3-AAV), MPO-ANCA-positive (MPO-AAV) or ANCA-negative. The latter encompasses patients with the clinical and pathologic features of ANCA vasculitis without detectable ANCAs.

There are several arguments to support this ANCA-based approach, namely the proven in vitro and in vivo pathogenicity of ANCAs and the significant correlation of specific ANCAs with both genetic backgrounds and clinical manifestations. The limitations of this approach, and of (over-)relying on ANCAs in general, include: (i) the phenomenon of secondary (“by-stander”) ANCAs in entities other than AAV but with similar clinical presentation; (ii) the possible discordance between serum ANCA titres and disease activity; and (iii) ANCA-negative patients with the clinical and pathologic features of ANCA vasculitis. The latter reportedly accounts for <5% of AAV, depending on the screening method (including the use of indirect immunofluorescence or not) and ELISA tests [1].

Although AAV may affect any organ during its course, there is a propensity for respiratory tract and renal involvement. The latter typically presents in the form of relapsing-remitting pauci-immune necrotizing and crescentic glomerulonephritis and carries the considerable risk of progression to end-stage renal disease (ESRD). It is thus essential to detect and manage renal involvement promptly due to its significant contribution to disease morbidity and mortality. However, we still lack specific, non-invasive biomarkers that would reflect disease activity and indicate underlying AAV-related renal inflammatory involvement. The established practice is thus to combine serum ANCA titres with the markers of kidney function and systemic inflammatory response and then proceed to renal biopsy, which is considered to be the diagnostic gold standard of renal involvement in AAV. Histologic parameters also provide a key input in the scoring systems that have been developed and validated to provide a prognostic value reflecting the progression to ESRD [2,3].

The etiopathogenesis of AAV is multifactorial and has not been fully elucidated, which translates to suboptimal disease management and considerable overall treatment-related morbidity and mortality [4]. In AAV pathogenesis, evidence of epigenetic influence has also accumulated, especially on a transcriptional level. Though altered DNA methylation patterns and histone modifications have already been convincingly implicated as epigenetic contributors to AAV pathogenesis, less clearer is the potential pathogenic role of microRNAs (miRNAs) in AAV [5,6]. These short, non-coding, single-stranded RNA molecules exert a post-transcriptional, mainly negative regulation of target messenger RNA (mRNA) translation. Specific expression profiles of certain miRNAs have thus been characterized and implied in pathogenesis. Furthermore, they are considered to be potential diagnostic biomarkers in several autoimmune diseases [7,8].

Comprehensive data on the expression alterations of miRNAs in AAV-affected renal tissue are presently lacking. Current knowledge regarding the role of miRNAs in AAV stems from studies that have produced contradictory data and employed a limited number of selected miRNAs to be analysed in biofluids, mainly serum. However, in the absence of tissue–serum correlation, it is difficult to establish whether these results reflect tissue-specific involvement in AAV or the vasculitic process in general. To our knowledge, intrarenal miRNA expression in AAV has not been comprehensively studied, especially in comparisons with both healthy adults without renal involvement and patients with glomerulonephrites other than AAV (GN). Such a disease- and tissue-specific miRNA signature could in turn reflect renal involvement in AAV. Using an extensive screening panel, we compared the intrarenal miRNA expression profiles in AAV with biopsy-proven renal involvement to subjects without the clinical presentation of renal disease (CTRL) and patients with GN in order to identify an AAV-specific renal tissue expression profile.

## 2. Results

### 2.1. Characterization of AAV and Control Group Cases

After isolation, reverse transcription (RT) and the quantification of reference miRNAs as suggested by the manufacturer (miR-16, miR-103a-3p, miR-191-5p, let-7a), a similar expression level among all samples was found in 26/30 AAVs, 10/15 CTRLs and 16/20 patients with GN initially considered for inclusion.

The AAV group thus comprised 26 (13 MPO-AAV and 13 PR3-AAV) patients with the following clinical characteristics: age 16–89 years (mean 62.3, SD 17.6), estimated glomerular filtration rate (eGFR as per the MDRD equation) < 90 mL/min in 25/26, ranging 6–86 mL/min (median 20, IQR 39) and daily proteinuria > 150 mg in 25/26, ranging 0.2–5.88 g (median 2.0, IQR 2.16). All AAV patients also presented with dysmorphic erythrocyturia as determined by the microscopic analysis of the urine sediment. The exact quantification of erythrocyturia using a counting chamber was available in 20/26 and ranged 31–6550 (median 502, IQR 1450) erythrocytes/1 µL native urine.

Detailed histopathological findings of the AAV group with established risk-stratifying scoring systems of AAV are presented in Table 1.

The control group included 26 patients, among which 10 belonged to CTRL, while there were 16 patients with GN. CTRL consisted of six pre-implantation living donor biopsies and four cases with an isolated thin basement membrane on electron microscopy without pathohistological change on light microscopy. Subsequently, we found out that one patient with a thin basement membrane presented with eGFR < 90 mL/min.

Patients with GN included three IgA nephropathies, four lupus nephrites (LN), four anti-GBM glomerulonephrites, three post-infectious glomerulonephrites and two membranous nephropathies.

The age in the control group as a whole ranged 11–78 years (mean 43.9, SD 17.3). eGFR was ≥90 mL/min (normal) in 13/26, while ranging 5–86 mL/min (median 26, IQR 53) in 13/26. Similarly, daily proteinuria was ≤150 mg in 7/26 and ranged 1–34.5 g (median 1.9, IQR 1.7) in 19 of the 26 included.

Interstitial fibrosis/tubular atrophy (IF/TA) % ranged 0–10 (median 0, IQR 5) in CTRL and 0–30 (median 0, IQR 5) in the patients with GN subgroup.

Subgroup-stratified clinical characteristics of AAV patients and controls are further presented and compared in Table 2.

### 2.2. Identification of AAV-Specific miRNAs

#### 2.2.1. Expression of miRNAs in AAV, Patients with GN and CTRL Groups

After examining all of the control reactions included in the miRNOME panel, we identified 372 miRNAs expressed in all eight pools of samples. The number of miRNAs that were not expressed in any pool was 93 and the number of miRNAs that was expressed in at least one pool but not in all eight was 288. However, there was no specific pattern of absence or presence of particular miRNAs in either disease or control samples. According to the manufacturer’s instructions, after eliminating all of the values above cycle 35, the observed correlation between the two RTs of the two pools of the same type (e.g., healthy control) was above 0.96. The data are presented in Figure 1A–E.

#### 2.2.2. Statistically Significant Difference in Expression

We additionally performed comparisons between AAV, CTRL and patients with GN. We observed 147 differentially expressed miRNAs between any of the two groups, with 84 between AAV and CTRL, 70 between patients with GN and CTRL and 35 between AAV and patients with GN. We thus found 29 miRNAs that statistically differentiated patients with AAV and patients with GN from CTRL and 17 miRNAs that discriminated AAV from both patients with GN and CTRL. Since the main focus of our research was AAV, we focused on those 17 so-called AAV-specific miRNAs in our further analyses. The results are summarized in Figure 2.

#### 2.2.3. Characterization of AAV-Specific miRNAs

Using the miRBase, we characterized the identified miRNAs and found that these 17 miRNAs belong to eight miRNA families and that they are located on 14 chromosomal loci on 12 chromosomes. Twelve of the miRNAs are clustered with other miRNAs and 13 might be part of other protein-coding or non-protein-coding genes that are transcribed in the same or opposite direction. The majority of identified miRNAs are from the leading strand of miRNA transcripts, while four are from passenger strands. We also found that the majority of miRNAs have single-nucleotide polymorphism (SNP) in their mature sequence; however, not all SNPs are located in the seed region. Regardless, in the light of non-canonical binding, the SNPs from the non-seed region might also be important for miRNA function. Their characterization and the known functions, processes and pathways implicated in AAV are summarized in Table 2. The heat map in relation to the CTRL group is summarized in Figure 3.

#### 2.2.4. Target Prediction and Annotation of Selected miRNAs

For the six miRNAs (hsa-miR-24-2-5p, hsa-miR-96-5p, hsa-miR-130b-3p, hsa-miR-376a-5p, hsa-miR-508-3p and hsa-miR-769-5p) without any publications that functionally relate them to the processes and pathways activated in AAV, we first searched across the miRTarBase and the TarBase. We were able to identify hsa-miR-24-2-5p in the miRTarBase and hsa-miR-24-2-5p, hsa-miR-96-5p, hsa-has-miR-130b-3p and hsa-miR-376a in the TarBase. The results are summarized in Table 3. For hsa-miR-508-3p and hsa-miR-769-5p, we predicted targets using TargetScan, miRDB and TargetMiner. The predicted transcripts were further analysed using DAVID for KEGG pathway involvement and tissue expression. The results are summarized in Table 4, Table 5 and Appendix A.

## 3. Discussion

The aim of this screening study was to discover and characterize the renal tissue expression profile of AAV-specific miRNAs in treatment-naïve patients in relation to both healthy controls and patients with GN. Importantly, the AAV sample pool comprised both MPO- and PR3-AAV cases, since the main goal of this study was to identify candidate miRNAs that differentiated AAV as a disease group from other (auto)inflammatory and non-inflammatory conditions, including healthy controls. By employing a broad screening panel, we identified 17 individual miRNAs whose expression levels, statistically speaking, differentiated AAV from both the healthy controls and the patients with GN sample pools. The most important findings include members of the miR-30 family, followed by miRNAs related to processes and cytokines previously deemed important in AAV pathogenesis, namely monocyte/macrophage polarization (hsa-miR-142-5p, hsa-miR-150-5p, hsa-miR-181a-5p, hsa-let-7a-5p), T-cell activation and differentiation (hsa-miR-542-5p) and related interleukin 6 receptor (IL-6R) signaling (hsa-miR-204-5p), the B-cell activating factor (BAFF, hsa-miR-30a-3p), renal fibrosis (hsa-miR-21-3p and hsa-miR-150-5p) and endothelial injury (hsa-miR-204-5p). Other processes and renal diseases in which the miRNAs differentially expressed in our screening panel have been implicated include miRNAs involved in acute allograft kidney rejection (hsa-miR-30a-3p, hsa-miR-142-5p and hsa-miR-150-5p) and lupus nephritis (LN) (hsa-miR-30b-5p and hsa-miR-150-5p). Finally, the screening identified seven miRNAs (hsa-miR-24-2-5p, hsa-miR-96-5p, hsa-miR-130b-5p, hsa-miR-376a-5p, hsa-miR-508-3p, hsa-miR-582-5p and hsa-miR-769-5p) that have not yet been affiliated with the pathogenesis of any non-neoplastic renal disease. Several miRNAs identified in our study have indeed been implicated in renal diseases other than AAV. However, this is understandable and acceptable given that a single miRNA may regulate a large number of different (unrelated) mRNA transcripts [20]. 

Five miR-30 family members, hsa-miR-30a/b/c/d/e, were expressed differentially in our screening model and were therefore possibly related to the pathogenesis of AAV. One such identified example was hsa-miR-30a-3p. Importantly, the BAFF mRNA transcript has been predicted to be a target of hsa-miR-30a-3p in human fibroblasts from rheumatoid arthritis and systemic sclerosis patients [10]. It has already been established that autoreactive B cells and ANCA-secreting plasma cells are prominently involved in AAV pathogenesis. Furthermore, a positive feedback loop involving BAFF secretion by ANCA-activated neutrophils results in increased B cell survival and antibody production [21]. If a regulation mechanism similar to that identified in the study of Holden et al. [21] also applied to neutrophils in AAV, it might underpin the feedback loop. Interestingly, miR-30 family members areotherwise considered to be podocyte enriched and are responsible for maintaining a homeostatic state due to their inhibitory effect on the injury-mediating Notch1 and p53 signaling pathways. Accordingly, marked downregulation of miR-30 family members has been observed in focal segmental glomerulosclerosis (FSGS), predisposing podocytes to apoptosis, calcineurin-induced actin meshwork remodeling, and culminating in foot process effacement and podocyte detachment [11,22]. Although renal involvement in AAV is not primarily considered to be a form of podocytopathy, functional podocyte defects, such as their detachment and podocyte density, have nonetheless been associated independently with both the recovery of renal function and ESRD risk in AAV [23]. hsa-miR-30a-3p has also been described as indicative of ongoing acute rejection in comparison to non-rejecting normal allograft renal biopsies [24]. Finally, downregulated hsa-miR-30b-5p has been involved in proliferating LN downstream of increased interferon-α (IFN-α), attesting to the pathogenic role of type 1 IFN-related signaling in this disease [12].

hsa-miR-142-5p, which is related to monocytes/macrophages, was identified as differentially expressed in the AAV pool during our screening. Monocytes and macrophages have also been increasingly implicated in AAV pathogenesis as the actual initiators and maintainers of the vasculitic process [25]. Specifically, an alternatively activated (M2-polarized) macrophage linked to a Th2-dominant (anti-inflammatory and pro-fibrogenic) response is considered to be the prevailing phenotype in early, established and quiescent AAV lesions and is further correlated with ESRD risk in one study [26,27]. In keeping with this, hsa-miR-142-5p has been demonstrated to promote and maintain M2 macrophage polarization by targeting suppressor of cytokine signaling 1 (SOCS1), resulting in sustained phosphorylation and signaling via the signal transducer and activator of transcription (STAT) 6, which is considered to be an essential transcription factor in M2-related gene expression [28,29].

Another miRNA identified as differentially expressed in the AAV pool in our study was hsa-miR-150-5p. The upregulated expression of this miRNA has also been observed in M2-polarized macrophages [14]. The additional association of hsa-miR-150-5p pertains to neutrophil-derived extracellular vesicles and specifically neutrophil-derived microvesicles (NDMV), which are released by neutrophils already present at the inflammation foci and which contain abundant hsa-miR-150-5p. Indeed, NDMVs have also been associated with inflammation suppression via M2-oriented macrophage polarization [30]. Alternatively, this finding could result from the expression of hsa-miR-150-5p by parietal epithelial cells, which constitute a prominent cell type in glomerular crescents, the prime active renal lesions in AAV [31]. Characterized in a Henoch-Schönlein purpura rat model, SH2-domain-containing tyrosine phosphatase 2 (SHP2) is another potential modulator of macrophage polarization due to its inhibitory effect on the M2 polarization-promoting STAT3 signaling pathway while also actively inducing M1 polarization. Conversely, SHP2 has been considered to be a plausible target of hsa-miR-181a-5p, which could result in SHP2 under-expression and thus the disinhibition of STAT3 signaling [15]. Indeed, hsa-miR-181a-5p was among those differentially expressed in our AAV sample pool.

Another identified miRNA in our study that related to monocytes was hsa-let-7a. An interesting feature observed in the monocytes in AAV is their increased expression of membrane-bound Integrin alpha M (CD11b) [32]. This translates to facilitated target tissue migration through the vascular endothelium and to pro-inflammatory macrophage polarization, possibly via the upregulated expression of the hsa-let-7a miRNA, which we also identified as differentially expressed. Furthermore, the upregulated expression of hsa-let-7a correlated inversely with IL-6 expression in human macrophages in a study by Schmid et al. [19]. In this regard, it is notable that IL-6 also enhances the STAT3-dependent polarization of macrophages towards the M2 phenotype [33]. It is thus tempting to speculate that monocytes primarily exert pro-inflammatory effects, while the tissue macrophages into which they differentiate, polarize towards the M2 phenotype, which could ultimately translate to chronic-type inflammation and result in fibrogenesis. These phenotypic shifts could in part be modulated by miRNAs, as discussed above.

An interesting observation pertained to candidate hsa-miR-582-5p, which was found to be upregulated in monocytes from patients with active tuberculosis, resulting in monocyte apoptosis inhibition through the targeting of forkhead box protein O1 (FOXO1) [18]. Contrary to neutrophils, monocyte apoptosis has not been considered to be pathogenetically significant in AAV in the existing literature. However, given their established pathogenic implications and their prominence in AAV lesions, it would be sensible to consider this aspect potentially important as well.

Candidate hsa-miR-542-5p and hsa-miR-204-5p have also been described in relation to aberrant T-cell activation and IL-6 signaling, which are processes considered to be important in AAV pathogenesis. Specifically, autoantigen-specific Th17 effector cells and defective regulatory T (T_reg_)-cells with a Th17-like phenotype have been implicated [34,35]. In a study by Pang et al., both Th17 and T_reg_ Treg differentiation from native T-cells and the ratio between these two T-cell subtypes have been related to the overexpression of hsa-miR-542-5p and its mice homologue (mmu-miR-322-5p) in myeloid-derived suppressor cells in progressive systemic lupus erythematosus [17]. Additionally, this study observed transforming growth factor-β (TGF-β) pathway activation because of mmu-miR-322-5p overexpression, which has been linked to the differentiation of naive T-cells to a Th17 phenotype [36]. Th17 cells in turn relate significantly to the process of neutrophil priming, since Th17-related cytokines in turn stimulate macrophages to produce priming-promoting factors, such as IL-1β and tumor necrosis factor α (TNF-α). IL-6/IL-6R signaling is also prominently involved in AAV, as evidenced by its established role in the neutrophil priming and monocyte activation process, IL-6-dependent Th17 and Treg differentiation, its elevated serum levels in AAV and, ultimately, the potential therapeutic value of anti-IL-6 antibody therapy [37,38]. Interestingly, the downregulation of hsa-miR-204-5p, otherwise considered to be abundantly present in unscarred and uninflamed renal parenchyma, is related to the elevated expression of IL-6R in human renal tubular epithelial cells in response to pro-inflammatory stimuli, including TNF-α and IL-1β [16]. It would be interesting to research whether such regulation applies to neutrophils in AAV, given that TNF-α, IL-1β and IL-6 are neutrophil-priming factors and that IL-6R is expressed in them. It would be important to see whether hsa-miR-204-5p also regulates IL-6R expression in naïve T-cells, which might influence their differentiation towards pathogenically relevant Th17 effector and Treg cells. Furthermore, miR-204 was thought to regulate the endothelial endoplasmic reticulum stress response and to be central to the development of endothelial dysfunction in human endothelium [39]. Consequently, the observed differential expression of this miRNA in our screening could in part relate to extensive endothelial dysfunction and damage, known to be present in AAV [40].

An interesting finding of this study was that two differentially expressed miRNAs (hsa-miR-21-3p and hsa-miR-150-5p) were strongly associated with the process of renal fibrosis, despite the fact that we actively sought to minimize the amount of chronic glomerular, vascular and tubulointerstitial change in both AAV patients and controls. The process of fibrosis is similar across different organs and encompasses the replacement of functional tissue with extracellular matrix components, the proliferation of fibroblasts in chronic inflammation and epithelial-to-mesenchymal transition [41]. A possible reason for the detection of fibrosis-related miRNAs among AAV-specific miRNAs in our study is that there was a slightly higher median % IF/TA in AAV compared to the CTRL and GN samples, although in both groups, fibrosis was minimal and focal at most.

In humans, existing knowledge of the potential involvement of miRNA in AAV with renal involvement is scarce and stems from either serum- or urine-based studies, some of which have been based on a limited screening panel or a set of pre-selected miRNAs [42,43,44,45,46,47]. Importantly, these studies did not include patients with GN other than AAV among control samples to enhance their specificity. The origin and significance of thusly defined miRNA is therefore unclear. A lack of consensus on the normalization of miRNA expression in biofluids is another issue to address and may well translate to incongruous results [48]. Additionally, correlating tissue and biofluid expression of miRNAs is also fraught with difficulties other than study design. An important point to raise here is that the expression of (renal) tissue-profiled miRNAs is frequently not faithfully reproduced in biofluids [49]. Surprisingly, however, none of the differentially expressed miRNAs identified by our screening panel on renal tissue matched any of the hitherto characterized AAV-related miRNAs.

A subset of AAV-associated miRNAs characterized by this and previous studies could reflect either secondary compensatory responses to the AAV-related inflammatory process or the bystander phenomenon, due to their genomic locations, and might reflect the consequences rather than causality in pathogenesis. Further, miRNAs are spatially restricted regulators of cellular populations with varying expression levels in different renal compartments. Thus, an important limitation of a renal biopsy-based approach is the confounding effect of tissue heterogeneity, since this might result in falsely identifying or missing the differential expression of region- or cell-specific miRNAs [50]. Additionally, it must be emphasized that the results obtained in this study originate from pooled samples-screening, and a validation study of these candidate differentially expressed miRNAs is therefore mandatory to establish their potential diagnostic utility.

## 4. Materials and Methods

### 4.1. Selection of Patients and Controls

This was a retrospective study conducted at the Institute of Pathology, Faculty of Medicine, University of Ljubljana. Selected AAV and control cases were retrieved from archived pathology reports of kidney needle biopsies performed in the years 2000–2020. For the purpose of this study, biopsies that contained at least seven glomeruli were considered to be representative. An experienced nephropathologist (N.K.) reviewed the selected cases prior to inclusion.

The AAV group included 15 treatment-naive MPO and 15 PR3-positive AAV patients with the histological verification of florid renal involvement in the form of pauci-immune crescentic glomerulonephritis and no significant renal (glomerular) co-pathology, with up to focal (<30%) IF/TA as determined in the renal cortex of the kidney biopsy. The histological classification of AAV was recorded [2] and ANCA Renal Risk Score (ARRS) [3] was calculated.

The control group included 15 samples that were considered to approximate normal, i.e., non-inflammatory, non-scarring kidney biopsy (CTRL). Control samples were collected from subjects with no symptoms/signs of kidney disease (i.e., living donors) or from subjects who had undergone kidney biopsy due to minimal isolated urine abnormalities (microhematuria and/or microalbuminuria) and normal renal function. In this context, eGFR as per the MDRD equation ≥90 mL/min/1.73 m^2^ and daily proteinuria values ≤ 150 mg/day (24-h urinary total protein excretion measured with colorimetric method (Auto Analyzer Olympus OSR6170, Tokyo, Japan) were acceptable for inclusion.

To increase the specificity of the study, the control group also included 20 patients with GN other than AAV, which consisted of immune complex and anti-GBM glomerulonephrites with or without necrotizing/crescentic features, focal (<30%) IF/TA and negative ANCAs by immune serological tests (Wieslab, Svar Life Science AB, Malmö, Sweden).

Age, gender, eGFR, daily proteinuria values (measured in 24 h urine collections), and IF/TA (% degree in 5% increments) at kidney biopsy were recorded for each included case. We additionally collected erythrocyturia values (semiquantitative urine sediment analysis and erythrocyte count in native urine with a Bürcker–Türk counting chamber (BRAND™ Bürcker Türk Counting Chambers, ThermoFisher Scientific, Foster City, CA, USA; reference values are ≤10/µL) in AAV patients.

We made every effort to match AAV patients and controls by age, gender, and IF/TA to the maximum extent possible.

### 4.2. RNA Isolation

For the isolation procedure, four 10 µm thick sections were cut from the formalin-fixed paraffin-embedded (FFPE) needle renal biopsy tissue blocks. Following deparaffinization using 1 mL of xylene (Sigma-Aldrich, Merck KGaA; Darmstadt, Germany), total RNA was isolated using the MagMAX™ FFPE DNA/RNA Ultra Kit (ThermoFisher Scientific; Foster City, CA, USA) according to the manufacturers’ instructions. The only modification included protease digestion overnight. RNA was eluted in 50 µL of nuclease-free water and stored at −80 °C. To assess the RNA purity and isolation yield, the concentration was measured using NanoDrop-One (ThermoFisher Scientific; Foster City, CA, USA) and Qubit 3.0 (Invitrogen, ThermoFisher Scientific; Foster City, CA, USA) with a high-sensitivity RNA assay.

### 4.3. Reverse Transcription and Quality Control of Resulting cDNAs

Using the miRCURY LNA RT Kit (Qiagen; Hilden, Germany), the reverse transcription of total RNA was subsequently performed according to the manufacturer’s instructions in a 10 μL reaction master mix with 10 ng of total RNA. The successful RT procedure was confirmed by adding spike-in RNAs and with the subsequent quantification of these spike-in RNAs (UniSp6). The resulting RT was diluted 60-fold and 3 μL was used in a 10 μL reaction master mix, according to the manufacturer’s instructions, to quantify the reference genes suggested by the manufacturer (miR-16, miR-103a-3p, miR-191-5p, let-7a). Quantitative real-time polymerase chain reaction (qPCR) was carried out using Rotor Gene Q (Qiagen; Hilden, Germany). All the qPCR reactions were performed in duplicate. Following amplification, a melting curve analysis of the PCR products was performed to verify the specificity and identity. Melting curves were acquired on the SYBR channel using a ramping rate of 0.7 °C/60 s for 60–95 °C.

### 4.4. Screening of miRNAs Expression by Quantitative Real-Time PCR (qPCR)

The screening was performed using an miRCURY LNA miRNA miRNOME panel (Qiagen; Hilden, Germany) that included 768 miRNA-specific and control oligonucleotides. Prior to proceeding with qPCR, eight separate cDNA pools were formed, two for each disease state (MPO-ANCA, PR3-ANCA, non-inflammatory, non-scarring controls and GN controls) using 40 ng of cDNAs per pool. Samples for pools were selected based on the results of the amplification of the UniSp6, *miR-16*, *miR-103a-3p*, *miR-191-5p* and *let-7a*. Mastermix for screening with cDNAs pool, QuantiNova SybrGreen and ROX was performed and loaded according to the manufacturer’s instructions. For each of the eight pools, separate screening was performed using QuantStudio 7 Pro (Thermo Fisher Scientific; Foster City, CA, USA). The signal was collected at the endpoint of every cycle. To ascertain the specificity of the qPCR products, a melting curve analysis was performed using a ramping rate of 0.075 °C/1 s for 60–95 °C.

### 4.5. Statistical Analysis of miRNA Expression

The normalization factor was calculated using the Cqs of an inter-plate calibrator according to the manufacturer’s instructions. After inter-plate calibration, the global mean expression of miRNA for each sample was calculated and used to normalize the expression of analysed miRNAs (ΔCt) [51]. Only miRNAs that were expressed in all samples were used for global mean expression. Differences in normalized individual miRNA expression values (ΔCt) were then tested for statistical significance between the replicates of four pools of samples using a t-test. We considered two-tailed test *p*-values < 0.05 to be statistically significant.

### 4.6. Annotation of Best Candidate miRNAs

For statistically significant differentially expressed miRNAs between AAV and CTRL and between AAV and patients with GN pools, we checked their reliability, location, family, clustering with other miRNAs and co-localization with other genes for potential intronic location in the miRBase [52]. We also checked for SNPs (miRNASNP-v3) located in genes for miRNAs [53]. Searching the literature, we checked for their proven involvement in renal diseases and/or renal tissue expression. For those that did not meet the last two criteria, we tried to further annotate them. Using online databases TarBase v.8 [54] and miRTarBase [55], we first analysed the remaining miRNAs. The inclusion criteria for target genes were based on a validation method with a low-throughput validation method in Tarbase v.8 and a strong validation method in miRTarBase. TarBase includes low- and high-throughput methods, with Western blot, qPCR and reporter assay being low-throughput and microarray being high-throughput. Similarly, miRTarBase distinguishes between strong evidence methods—reporter assay, Western blot and qPCR—and less strong validation methods, such as microarray, next-generation sequencing (NGS) and pSILAC. For those that were present in these two databases, we searched for the annotated KEGG pathways. For others, we predicted targets using TargetScan, miRDB and TargetMiner, i.e., tools that are included within the miRBase [56,57] and annotated predicted targets using the DAVID tool [58,59].

## 5. Conclusions

The fact that a substantial proportion of the differentially expressed miRNAs might be related to established biological processes that are considered to be important in AAV pathogenesis, such as macrophage polarization, T-cell differentiation associated with IL-6/IL-6R signaling, B-cell survival and BAFF signaling, renal fibrosis and endothelial injury, lends credence to the results presented here. However, a validation study of these differentially expressed miRNAs in an independent, larger sample cohort is mandatory to establish their potential diagnostic utility.

## Figures and Tables

**Figure 1 ijms-23-00105-f001:**
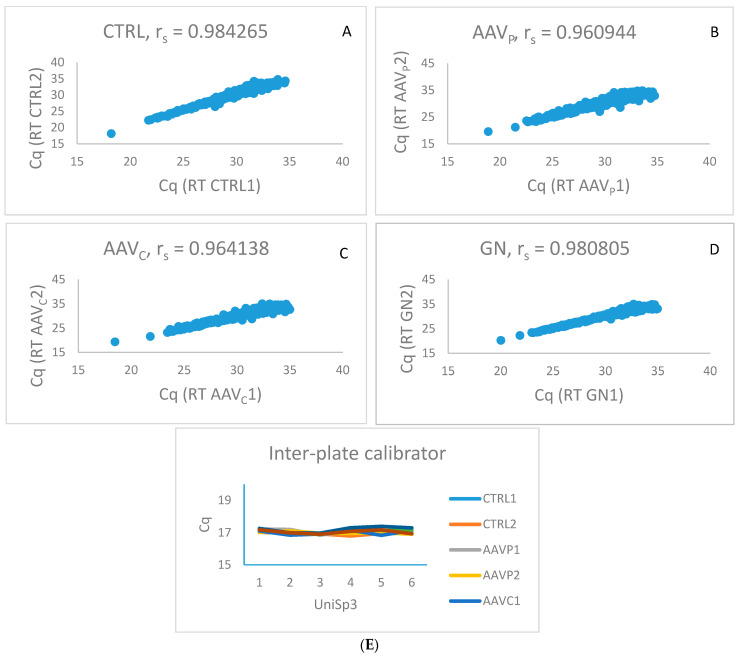
Quality control of the results of analysed sample pools. (**A**–**D**). Correlation of expressions of miRNAs between two pools of the same group of samples: (**A**) correlation between two pools of healthy controls (CTRL); (**B**,**C**) correlation between four pools of ANCA-associated vasculitis (AAV); (**D**) correlation between two pools of glomerulonephrites (GN); (**E**) Expression of inter-plate calibrators in analysed pools of samples. Six inter-plate calibrators were located on two miRNOME panels (miRNOME panel I in miRNOME panel II).

**Figure 2 ijms-23-00105-f002:**
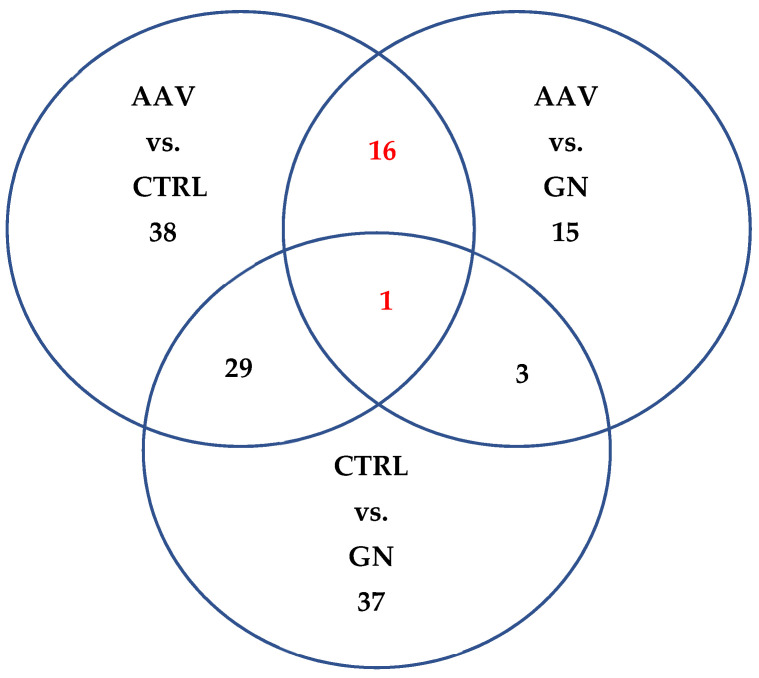
Venn diagram of the number of statistically significant differentially expressed miRNAs. Legend: AAV, ANCA-associated vasculitis; CTRL, subjects without clinical presentation of renal disease; GN, patients with glomerulonephrites other than AAV.

**Figure 3 ijms-23-00105-f003:**
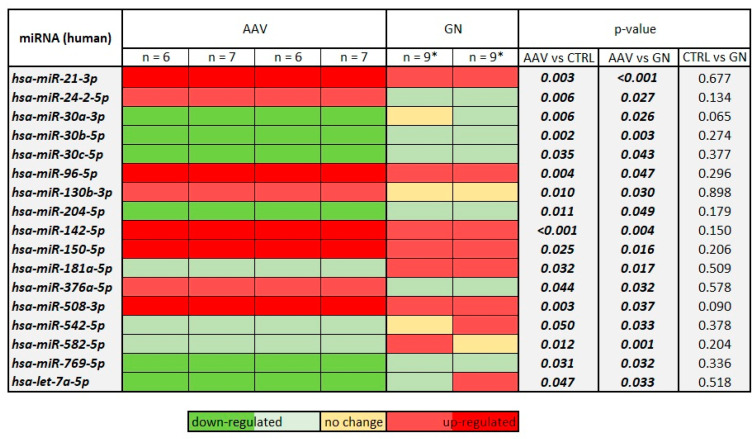
Expression heat map of 17 AAV-specific miRNAs in AAV and GN in comparison to controls without the clinical presentation of renal disease. Legend: AAV, ANCA-associated vasculitis; GN, patients with glomerulonephrites other than AAV; n, number of samples included in the pool; *, two samples were included in both pools of GN samples.

**Table 1 ijms-23-00105-t001:** Histopathological characteristics of the AAV group as a whole and by ANCA serotype.

	AAV Histological Classification ^+^	% Normal Glomeruli	% Active Glomerular Lesions *	% Global Glomerular Sclerosis	% IF/TA	ARRS Group
AAV	F: 5/26 (19%) C: 15/26 (58%) M: 6/26 (23%) S: 0/26 (0%)	24 (0–90, 36)	54 (7–86, 45)	8 (0–36, 11)	10 (0–30, 6)	Low: 9/26 (35%) Medium: 12/26 (46%) High: 5/26 (19%)
MPO	F: 3/13 (24%) C: 5/13 (38%) M: 5/13 (38%) S: 0/13 (0%)	31 (0–90, 31)	30 (7–67, 43)	13 (0–36, 13)	10 (0–30, 15)	Low: 5/13 (38%) Medium: 6/13 (46%) High: 2/13 (16%)
PR3	F: 2/13 (15%) C: 10/13 (77%) M: 1/13 (8%) S: 0/13 (0%)	16 (8–59, 27)	71 (25–86, 29)	5 (0–15, 9)	10 (0–15, 5)	Low: 4/13 (31%) Medium: 6/13 (46%) High: 3/13 (23%)

Legend: ^+^ as per Berden et al. [2] (F—focal, C—crescentic, M—mixed and S—sclerotic classes); * % cellular/fibrocellular crescents with/without glomerular fibrinoid necrosis; IF/TA, interstitial fibrosis/tubular atrophy; ARRS, ANCA renal risk score [3] for progression to end-stage renal disease; all % values are expressed as median (range, IQR).

**Table 2 ijms-23-00105-t002:** Demographic and clinical characteristics of patients and controls by subgroups.

	M:F Ratio	Age ^+^	eGFR Normal	eGFR if <90 mL/min *	DP Normal	DP if >150 mg *
**AAV**	**14:12**	**62.3 (17.6)**	**1/26**	**20 (39)**	**1/26**	**2.0 (2.16)**
MPO	4:9	67.8 (15.2)	0/13	20 (29)	0/13	2.4 (3.8)
PR3	10:3	56.7 (18.6)	1/13	22 (48)	1/13	2.0 (1.9)
**CONTROL**	**11:15**	**43.9 (17.3)**	**13/26**	**26 (53)**	**7/26**	**1.9 (1.7)**
CTRL	6:4	49 (19.0)	9/10	N/A	7/10	N/A
GN	5:11	40.7 (16.0)	4/16	26 (40)	0/16	2.0 (1.7)
*p*-Value	0.579 ^a^	<0.001 ^b^	0.001 ^a^	0.406 ^c^	0.05 ^a^	0.228 ^c^

Legend: CTRL, subjects without clinical presentation of renal disease; DP, daily proteinuria value in g/day (measured in 24 h urine collection); eGFR, estimated glomerular filtration rate in ml/min/1.73 m^2^ according to MDRD formula; GN, patients with glomerulonephrites other than AAV; M:F, male-to-female ratio; N/A, not applicable; *p*-values were calculated between AAV and CONTROL groups (bolded values); ^+^ mean (SD): * median (IQR); ^a^ Fisher’s exact test; ^b^
*t*-test, ^c^ Mann–Whitney test.

**Table 3 ijms-23-00105-t003:** Characterization of AAV-specific miRNAs.

MicroRNA	Chromosomal Location ^a^	MicroRNA Family	Clustered ^a^	Same Chromosomal Region/Intronic ^a^	No. of SNPs in miRNAs	Known Function ^b^	References
*hsa-miR-21-3p*	17q23.1	no	no	*VMP1, LOC1148227848*	7	TGF-β/Smad3 signaling (fibrosis)	[9]
*hsa-miR-24-2-5p*	19p13.12	miR-24	*miR-23a, miR-27a*	*MIR23AHG*	4	/	/
*hsa-miR-30a-3p*	6q13	miR-30	*miR-30c/e*	no	14	BAFF Notch1, p53 signaling (podocyte injury)	[10,11]
*hsa-miR-30b-5p*	8q24.22	miR-30	*miR-30d*	*LOC102723694*	1	IFN-α signaling (mesangial proliferation in LN) Notch1, p53 signaling (podocyte injury)	[11,12]
*hsa-miR-30c-5p*	1: 1p34.2 2: 6q13	miR-30	*1: miR-30e* *2:* no	*1: NFYC* *2:* no	1: 3 2: 7	Notch1, p53 signaling (podocyte injury)	/
*hsa-miR-96-5p*	7q32.2	no	*miR-182, miR-183*	no	7	/	/
*hsa-miR-130b-5p*	22q11.21	miR-130b	*miR-301b*	*LOC107985532*	6	/	/
*hsa-miR-142-5p*	17q22	no	*miR-4736*	*LOC111822952* (opposite direction)	4	SOCS1/STAT6 signaling (macrophage polarization)	[13]
*hsa-miR-150-5p*	19q13.33	no	*no*	no	6	PU.1 transcription factor (macrophage polarization)	[14]
*hsa-miR-181a-5p*	1: 1q32.1 2: 9q33.3	no	*1: miR-181b-1* *2: miR-181b-2*	*1: MIR181A1HG* *2: MIR1812HG, NR6A1*	1: 2 2: 7	SHP2/STAT3 signaling (macrophage polarization)	[15]
*hsa-miR-204-5p*	9q21.12	miR-204/211	*no*	*TRPM3*	4	IL-6 receptor (chemokine generation in renal tubular epithelium)	[16]
*hsa-miR-376a-5p*	1, 2: 14q32.31	miR-376	*1, 2: miR-300, miR-376b, miR-376c,* *miR-381, miR-487b, miR-495, miR-539, miR-543, miR-544a, miR-654, miR-655, miR-889, miR-1185-1, miR-1185-2*	no	5	/	/
*hsa-miR-508-3p*	Xq27.3	miR-606	*miR-506, miR-507*	*LOC105373347* (opposite direction)	2	/	/
*hsa-miR-542-5p*	Xq36.3	no	*miR-424, miR-450a-1, miR-450a-2, miR-450b, miR-503*	no	13	TGF-β signaling (Th17 and T_reg_ differentiation)	[17]
*hsa-miR-582-5p*	5q12.1	no	*no*	*PDE4D*	7	FOXO1 (monocyte apoptosis)	[18]
*hsa-miR-769-5p*	19q13.32	no	*no*	*PGLYRP1* (opposite direction)	4	/	/
*hsa-let-7a-5p*	1: 9q22.32 2: 11q24.1 3: 22q13.31	let-7	*1: let-7f-1, let-7d* *2: miR-100, miR-10526* *3: let-7b, miR-4763*	*1: MIRLET7A1HG, LINC02603* (opposite direction) *2: MIR100HG* *3: MIRLET7BHG*	1: 2 2: 4 3: 1	CD11b signaling (macrophage polarization)	[19]

Legend: SNP, single-nucleotide polymorphism; TGF-β, transforming growth factor-β; BAFF, B-cell activating factor; IFN-α, interferon-α; LN, lupus nephritis; SOCS1, suppressor of cytokine signaling 1; STAT, signal transducer and activator of transcription; PU.1, Purine-rich box-1 transcription factor; SHP2, Src homology region 2-containing protein tyrosine phosphatase 2; IL-6, interleukin 6; FOXO1, forkhead box protein O1; CD11b, integrin alpha M; ^a^ there are several copies of certain miRNAs within the genome, therefore the number represents the particular copy; ^b^ affiliation to signaling pathways or messenger RNA targets, while related biological processes are denoted in brackets.

**Table 4 ijms-23-00105-t004:** Target identification for four AAV-specific miRNAs.

MicroRNA	miRTarBase	TarBase			
		No	Strong Evidence Method	No. of Low Throughput Experiments	KEGG Pathways (Number, AAV Related)
*hsa-miR-24-2-5p*	*SPRY2, BCL2*	265	*SPRY*	1	6, none involved in kidney disease
*hsa-miR-96-5p*	/	902	*CTDSP1, FOXO1, SCAB1, CTNND1, CDH1, SNAI2, ZEB1*	10	27, EMT related (adherents function, ECM-receptor interaction, focal adhesion)
*hsa-miR-130b-5p*	/	1058	*PTEN, SMAD4, TGFBR2, TP63, ZBTB4*	3	12, fibrosis related (TGFB signaling)
*hsa-miR-376a-5p*	/	970	*ALK7*	1	6, fibrosis and immune response related (NF-kB signaling)

Legend: AAV, ANCA-associated vasculitis; ECM, extracellular matrix; EMT, epithelial-to-mesenchymal transition.

**Table 5 ijms-23-00105-t005:** Target prediction for two AAV-specific miRNAs.

MicroRNA	Prediction Method	Number of Predicted Targets	DAVID Tissue Expression of Predicted Targets	DAVID KEGG Pathways of Predicted Targets
*hsa-miR-508-3p*	TargetScan	2474	Endothelial cells (*n* = 12)	Focal adhesion (*n* = 36); NF-κB (*n* = 20), TCR (*n* = 19) and BCR signaling (*n* = 13)
	miRDB	417	Renal cell carcinoma (*n* = 6)	Renal cell carcinoma (*n* = 5)
	TargetMiner	458	Fetal kidney (*n* = 16)	FoxO signaling (n = 9); focal adhesion (*n* = 11); inflammatory mediator (*n* = 6)
*hsa-miR-769-5p*	TargetScan	3669	Kidney epithelium (*n* = 12)	TGFB (n = 29) and TNF (*n* = 12) signaling
	miRDB	297	/	TGFB signaling (*n* = 5)
	TargetMiner	493	Fetal kidney (*n* = 12), T-cell (*n* = 15), lymphocyte (*n* = 6)	TGFB signaling (*n* = 6); FoxO signaling (n = 10); focal adhesion (*n* = 12); renal cell carcinoma (*n* = 5)

Legend: BCR, B-cell receptor; TCR, t-cell receptor; TGFB, transforming growth factor beta; TNF, tumor necrosis factor.

## Data Availability

Data will be submitted upon request.

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
