# Peer review of "Renal Tissue miRNA Expression Profiles in ANCA-Associated Vasculitis—A Comparative Analysis"

_ijms, 2021, doi:10.3390/ijms23010105_

Round 1

Reviewer 1 Report

The present paper entitled "Renal tissue miRNA expression profiles in ANCA-associated vasculitis – A comparative analysis" by Matic Bošnjak and colleagues describe the alteration of miRNAs expression in ANCA-associated vasculitis by applying a miRCURY LNA miRNA miRNOME panel.

The authors identified several dysregulated miRNAs in ANCA-associated vasculitis group and characterized the role of these miRNAs by using the miRBAse database.

The research objective is original; however the manuscript needs to be improved in some points. 

Major:

The main issue is concerning the bioinformatic prediction of miRNAs targets. This prediction analysis  should be performed using at least three different bioinformatic prediction tools as miRANDA, miRBase, HumanScan, etc. (the authors just used TargetScan). I suggest the authors to remake this analysis with the aim to confirm/exclude/add findings.

Minor:

I suggest to revise the "2.1. Characterization of AAV and control group cases" section. Here, the authors could report just the considerations regarding the miRNAs dysregulation found in each group of patient.  The clinical features of patients can be collected in a "case selection" paragraph, in  Material and Method section.

Author Response

Point 1: The main issue is concerning the bioinformatic prediction of miRNAs targets. This prediction analysis  should be performed using at least three different bioinformatic prediction tools as miRANDA, miRBase, HumanScan, etc. (the authors just used TargetScan). I suggest the authors to remake this analysis with the aim to confirm/exclude/add findings.

Response 1: We appreciate your suggestion. However, since we were not able to find “HumanScan” and “miRANDA” was unavailable, we focused on two things, the miRBase and a review of available prediction programs. Although publications from 2020 listed a number of prediction tools, the majority of them are not functional. Besides miRNANDA, which is not available, we therefore used the other three prediction tools from the miRBase (TargetScan, miRDB and TargetMiner). The resulting changes are incorporated in “Materials and methods” and “Results (Table 4 and Supplementary Tables S1-S3)” sections. 

Point 2: I suggest to revise the "2.1. Characterization of AAV and control group cases" section. Here, the authors could report just the considerations regarding the miRNAs dysregulation found in each group of patient.  The clinical features of patients can be collected in a "case selection" paragraph, in Material and Method section.

Response 2: The authors thank Reviewer 1 for bringing up this suggestion. However, akin to miRNA expression levels, we consider that clinical and histological features better fit under Results section since they already reflect the data gathered as conceptualized in “4.1 Selection of patients and controls”. Additionally, we performed some statistical analyses which are presented together with this data and which we also consider results. Therefore, we would be inclined to retain the current format.

Reviewer 2 Report

In the present study, Bošnjak et al. describe miRNA expression in renal tissue miRNA comparing ANCA-associated vasculitis with other forms of GN and control samples. The study is interesting and provides further insights into potential miRNAs specifically attributed to ANCA GN. I recommend the following modifications to further strengthen the conclusions:

  1. Detailed information about the cohorts including ANCA GN should be provided. In detail, there should be information about ANCA GN class and a more thorough description of histopathological findings (crescents, sclerosis, necrosis, IF/TA, hematuria).
  2. Figure 2 needs labelling of color codes. In addition, meaning of individual boxes (4x AAV, 2x GN) remains unclear. The authors studied 26 ANCA GN, 16 other GN and 10 control patients. I recommend reporting individual datasets in each group and statistical evaluation. 
  3. Supplemental data is not available for the reviewer. Particularly, Supplemental Figures S1 and S2 include the screening data. I recommend to report these findings in the main manuscript (e.g. as heat map), because all further experiments are based on these results. 

Author Response

Point 1: Detailed information about the cohorts including ANCA GN should be provided. In detail, there should be information about ANCA GN class and a more thorough description of histopathological findings (crescents, sclerosis, necrosis, IF/TA, hematuria).

Response 1: Detailed information regarding the AAV cohort as a whole and further stratified by ANCA serotype has been included in a table form (Table 1) with a thorough presentation of suggested renal biopsy findings and established histopathologic classes/scoring systems. Numerical data are presented as % in the form of median value, followed by range and IQR in parentheses. Accordingly, we have omitted some text as to avoid redundancy and duplication.

Regarding erythrocyturia, the result of microscopic urine sediment analysis, which estimates the number of erythrocytes at x400 magnification (normal values in our laboratory are up to 3 erythrocytes/400x field), was available in all patients immediately before the renal biopsy. All patients included in the study had a large number of dysmorphic erythrocytes in the urine sediment. Since our laboratory does not give an exact numerical value when the number of erythrocytes in HPF is high (it only indicates very numerous erythrocytes in such cases), we decided not to report these data. However, we subsequently reviewed the results of microscopic counting of erythrocytes in urine by the Burcker-Turcker method (number of erythrocytes in 1 µL of non-centrifuged urine), which were available in 20/26 AAV patients. This is now included in text form under “2.1 Characterization of AAV and control group patients”.

Point 2: Figure 2 needs labelling of color codes. In addition, meaning of individual boxes (4x AAV, 2x GN) remains unclear. The authors studied 26 ANCA GN, 16 other GN and 10 control patients. I recommend reporting individual datasets in each group and statistical evaluation.

Response 2: According to reviewer’s suggestion, we have added colour codes and a statistical evaluation with an explanation of the meaning of individual boxes (each box is the pool of n number of samples and compared to the mean value of two pools of control samples, we also added the n). This figure is now labelled “Figure 3”.

Point 3: Supplemental data is not available for the reviewer. Particularly, Supplemental Figures S1 and S2 include the screening data. I recommend to report these findings in the main manuscript (e.g. as heat map), because all further experiments are based on these results.

Response 3: As suggested, we reported the screening data findings in a new figure, now labelled  “Figure 1 (A-E)” in the main manuscript and corrected the manuscript accordingly.

Round 2

Reviewer 1 Report

The authors replied to the comments 

Reviewer 2 Report

The authors addressed all my comments, I recommend acceptance of the manuscript in its current version.